# Navigating Challenges and Opportunities in Multi-Omics Integration for Personalized Healthcare

**DOI:** 10.3390/biomedicines12071496

**Published:** 2024-07-05

**Authors:** Alex E. Mohr, Carmen P. Ortega-Santos, Corrie M. Whisner, Judith Klein-Seetharaman, Paniz Jasbi

**Affiliations:** 1Systems Precision Engineering and Advanced Research (SPEAR), Theriome Inc., Phoenix, AZ 85004, USA; mohr@therio.me (A.E.M.); ortega@therio.me (C.P.O.-S.); cwhisner@asu.edu (C.M.W.); judith.klein-seetharaman@asu.edu (J.K.-S.); 2College of Health Solutions, Arizona State University, Phoenix, AZ 85004, USA; 3Biodesign Institute Center for Health Through Microbiomes, Arizona State University, Tempe, AZ 85281, USA; 4Department of Exercise and Nutrition Sciences, Milken Institute School of Public Health, George Washington University, Washington, DC 20052, USA; 5School of Molecular Sciences, Arizona State University, Tempe, AZ 85281, USA

**Keywords:** precision medicine, systems biology, digital twin, artificial intelligence, machine learning, blockchain

## Abstract

The field of multi-omics has witnessed unprecedented growth, converging multiple scientific disciplines and technological advances. This surge is evidenced by a more than doubling in multi-omics scientific publications within just two years (2022–2023) since its first referenced mention in 2002, as indexed by the National Library of Medicine. This emerging field has demonstrated its capability to provide comprehensive insights into complex biological systems, representing a transformative force in health diagnostics and therapeutic strategies. However, several challenges are evident when merging varied omics data sets and methodologies, interpreting vast data dimensions, streamlining longitudinal sampling and analysis, and addressing the ethical implications of managing sensitive health information. This review evaluates these challenges while spotlighting pivotal milestones: the development of targeted sampling methods, the use of artificial intelligence in formulating health indices, the integration of sophisticated *n*-of-1 statistical models such as digital twins, and the incorporation of blockchain technology for heightened data security. For multi-omics to truly revolutionize healthcare, it demands rigorous validation, tangible real-world applications, and smooth integration into existing healthcare infrastructures. It is imperative to address ethical dilemmas, paving the way for the realization of a future steered by omics-informed personalized medicine.

## 1. Introduction

Multi-omics, an integrative approach within the systems biology framework, is an innovative field that combines various ‘omics’ technologies to concurrently evaluate multiple strata of biological data. This approach encompasses the synergistic analysis of genomics, transcriptomics, proteomics, and metabolomics, among other omics fields (e.g., microbiomics), employing an array of bioinformatics tools to gain a comprehensive understanding of complex biological systems [1]. With its first indexing in the National Library of Medicine in 2002 (PubMed search “multi-omics”; 31 December 2023), recent years have seen a rapidly increasing interest in the application of multi-omics alongside individual omics layers, more than doubling scientific publication numbers (2022–2023 *n* = 7390 vs. 2002–2021 *n* = 6345; Figure 1). Moreover, large funding governmental bodies recognize the potential for personalized health insights and refining diagnostics, as exemplified by the National Institutes of Health’s ‘Multi-Omics for Health and Disease Consortium’ [2]. The potential benefits of robust multi-omics pipelines are plentiful [3]. They may provide a deep understanding of disease-associated molecular mechanisms [4], facilitate precision medicine by accounting for individual omics profiles [5], foster early disease detection and prevention [6], aid in the discovery of biomarkers crucial for diagnosis, prognosis, and treatment monitoring [7], and spotlight molecular targets for innovative drug development or the repurposing of existing therapeutics [8,9].

While multi-omics offers substantial promise, its research and large-scale application in healthcare present several challenges. The process of cohesively integrating and normalizing data across varied omics platforms and experimental methods remains difficult. Additionally, due to the sheer volume and high dimensionality of multi-omics datasets, there is an imperative for sophisticated computational utilities and stringent statistical methodologies to ensure accurate data interpretation [1]. Recognizing individual variability in omics profiles and discerning their responses to various interventions are pivotal. Given the distinct sensitivities of biomolecules across the multi-omics chain, determining optimal sampling strategies within a context-dependent omics hierarchy becomes critical for extracting meaningful patient insights. Precision analysis methods, such as comparing patient data against their in silico counterparts, may offer an avenue for enhanced health understanding. Finally, ethical and regulatory aspects cannot be understated when discussing deep phenotyping. Employing multi-omics in personal health assessments presents several ethical issues, including data privacy, informed consent, and equal access to healthcare resources. As international services increase from diverse companies, safeguarding individual privacy and ensuring responsible data management practices are of prime importance.

Here, we review the challenges ahead and identify the pivotal elements across the translational spectrum, essential for generating impactful deliverables in the form of provider tools and novel diagnostics and treatments. We conceptualize key milestones that may unlock the potential of multi-omics assessments in personalized healthcare. These include addressing diverse sampling types, harnessing the potential of artificial intelligence (AI) for forming health indices [10], adopting advanced statistical modeling via digital twins (DTs), and utilizing blockchain technology to fortify data integrity and ownership [11]. While multi-omics holds immense promise for revolutionizing healthcare, successful implementation requires rigorous validation, real-world application, and seamless integration into existing healthcare systems. Addressing these challenges and ethical considerations is vital to ensure that omics-driven, personalized medicine is realized within the healthcare landscape.

## 2. Multi-Omics Layering: Considering a Realistic Hierarchy of Testing and Sample Collection Frequency/Timing in Precision Medicine

In precision medicine, understanding the dynamics of different omics layers is crucial. Across the multi-omics spectrum, a diverse array of system layers informs biological analyses. In precision medicine, using a longitudinal model shows that not all omics layers follow the same sampling frequency. Indeed, understanding how often and when samples are taken in detailed biological characterization, or phenotyping, is not a new concept. For instance, Chen et al. (2012) found that certain omics layers, such as the transcriptome, shift dynamically from a healthy state to conditions such as viral infections or diabetes [12]. Identifying the most responsive omics layers is challenging because of the vast amounts of data generated in studies such as the one by Chen et al., where over three billion measurements were collected across 20 time points for just one participant. While the order of initial sampling may be relatively straightforward and subject-specific, a generally rational approach proposed by Hasin et al. (2017) for disease state phenotyping includes the genome, epigenome, transcriptome, proteome, metabolome, and microbiome [3]. While it is not the intent of this review to provide an in-depth discussion for each layer, we will provide an overview of their utility within a high-frequency, longitudinal framework and how sampling might better be approached in the context of precision medicine.

The genome, foundational in omics, is pivotal in creating a detailed profile of a patient’s molecular identity. It unveils specific genetic variations or mutations influencing genetic predispositions, inherited traits, susceptibility to diseases, and medication response. For example, cytochrome P450 enzymes demonstrate how genetic data can predict drug metabolism, enabling tailored treatment regimens [13]. While the genome gives a static snapshot of one’s genetic makeup, the epigenome, which represents changes in gene activity not involving underlying DNA sequences, is more dynamic. Integrating genetic data with other omics layers, such as transcriptomics and metabolomics, allows for a more comprehensive understanding of how genetic variations manifest at the molecular and metabolic levels. While the genome serves as a foundational starting point in omics, it anchors the flow of information, helping to pinpoint the origins of pathologies—whether genetic, environmental, developmental, and so on [3]. By integrating various omics layers, researchers can more confidently detect and interpret signals, identify patterns and co-occurrences, and replicate studies to robustly determine effect sizes.

The responsiveness to treatment and the frequency and collection timing (e.g., AM vs. PM) of assessments can vary based on the specific disease, treatment modality, and underlying molecular processes (Figure 2). The transcriptome, representing RNA molecules derived from genomic DNA in specific cells or tissues, informs gene expression dynamics. The transcriptomic layer is markedly sensitive to factors such as treatment, environment, and health behaviors, often necessitating more regular assessments relative to other omics layers [14,15,16]. For instance, studies of nightshift workers over a few days revealed significant changes in gene expression rhythms and desynchrony between rhythmic transcripts, shifting the sleep/wake cycle [17]. Approximately 3.0% of the human transcriptome showed up-regulation or down-regulation during the night shift condition. In certain cases, such as patient responsiveness to drug therapy, change may take months to years to manifest [18,19]. Moreover, the transcriptome is cell-specific, adding further complexity [20]. Despite the inherent complexities, consistent and personalized gene expression evaluations present several advantages [21,22,23]. Namely, they: (1) enable the identification of treatment-specific patterns, elucidating molecular action mechanisms, (2) serve as invaluable tools for biomarker discovery, capturing treatment-induced changes pivotal for prognosis and monitoring, (3) bolster pharmacogenomic research by providing insights into gene expression disparities that influence drug metabolism, targets, and transporters—augmenting drug efficacy while minimizing adverse events, and (4) facilitate the monitoring of treatment response, highlighting early indicators of treatment resistance or recurrence. However, it is essential to recognize that the choice of the most responsive and frequently assessed omics layer may vary depending on the specific disease context [23]. In some cases, other layers, such as proteomics or metabolomics, may provide more immediate and actionable information. 

Proteomics, a significant branch of multi-omics, focuses on studying proteins comprehensively, including their expression levels, post-translational modifications (PTMs), interactions, and functions within a biological system [24]. Moreover, proteomics plays a crucial role in understanding cellular processes, disease mechanisms, and treatment responses. By contrasting the protein expression profiles of healthy individuals with those afflicted by disease, or by comparing pre- and post-treatment states, proteomics can pinpoint differentially expressed proteins. Such proteins hold promise as potential biomarkers for disease diagnosis, prognosis, and therapeutic efficacy [23]. PTMs, central to protein functionality and cellular signaling, can be identified and quantified using proteomics, unraveling complex regulatory networks involved in various tissues and diseases [25,26]. Integrating proteomic data with other omics layers offers a comprehensive understanding of how genetic variations and gene expression changes influence protein activity and disease phenotypes. Moreover, proteomics plays a key role in identifying and validating potential drug targets, evaluating treatment effectiveness, and contributing to precision medicine by understanding disease heterogeneity and customizing treatments to individual characteristics [27,28].

Proteomic testing frequency is contingent upon the research or clinical context, the specific disease under investigation, and the objectives of the analysis. Unlike genomics, transcriptomics, and metabolomics, proteomics generally requires a lower testing frequency. This is attributed to the stability of proteins, which have longer half-lives compared to RNA or metabolites, making protein expression levels and modifications relatively stable over time [23]. Additionally, proteomic testing often involves complex workflows and sample availability challenges, as it requires tissue or biofluid collection [29]. To complement the overall multi-omics analysis, the testing frequency of proteomics should be aligned with specific clinical requirements and patient objectives. This may involve targeted time points, treatment interventions, or coordinated assessments with other omics data.

Metabolomics, a fundamental component of the systems biology paradigm, offers highly sensitive and variable data, making it essential to target more frequently in certain contexts. Metabolomics involves the study of small molecules (molecular weight < 2000 Da) implicated in diverse cellular functions and metabolic pathways [30]. These molecules can serve as direct indicators of cellular function and provide insights into the metabolic changes associated with disease and therapeutic responses [31,32]. Unlike other layers, such as the genome, which suggests potential cellular events, or the proteome, which identifies the agents behind these events, metabolomics grants a real-time perspective of ongoing metabolic activities [30]. Consequently, due to the rapid dynamics of metabolites, especially in response to therapeutic interventions, their assessment demands heightened frequency. Indeed, metabolic pathways might undergo modifications within mere minutes to hours post-treatment, reflecting the immediate impact of the treatment on cellular metabolism [33]. Importantly, metabolomic data capture the end products and intermediates of cellular metabolism. These metabolite signatures can reflect the integration of various biological processes, including gene expression, enzyme activity, and environmental factors. Highly valuable to precision medicine and integration within a multi-omics framework, metabolomic data can capture inter-individual and intra-individual variability, allowing for personalized treatment monitoring. By assessing metabolite profiles, clinicians can monitor individual responses to treatment, identify metabolic adaptations or changes, and make adjustments to optimize therapy.

The microbiome, characterized by its diverse microorganisms and intricate interactions [34], is a complex and unique layer in the multi-omics spectrum, making it challenging to place within the traditional (i.e., host) systems biology framework. The microbiome clearly has a profound impact on human health, functioning as an ancient and vital mutualistic connection [35]. The microbiome has pivotal roles in numerous diseases, with its composition and interactions heavily influenced by lifestyle determinants such as diet [36,37,38]. Focusing on the gut microbiome (GM), with a combined biomass of only ~200 g [34], it has been estimated that approximately half of all metabolites in the body have been synthesized or modified by gastrointestinal microbes [39]. Therefore, the assembly of microbes in our gut is tightly intertwined with our health and represents a potentially important therapeutic target. Due to its complex and diverse nature, the microbiome can be considered a “black box” of information. Advancements over the last several decades in DNA sequencing have made the microbiome more accessible for analysis, though many misconceptions are present in the current literature, forming a roadblock for scientific consensus and firming up a sound evidence base [40]. Regardless, an individual’s GM composition still holds great potential for providing valuable insights into gut health, nutrient metabolism, immune function, and associations with various health conditions.

The microbiome serves as a critical connector and communicator, interacting with various omics layers such as the genome, proteome, transcriptome, and metabolome to offer a comprehensive view of human health and disease. For example, at the genomic level, the host genome influences the composition and function of the microbiome [41]. Genetic variations in the host impact immune responses and mucosal surfaces, affecting the abundance and diversity of microbial species [42]. The microbiome’s influence on host gene expression occurs through mechanisms such as microbial metabolites, signaling molecules, or immune modulation, with the integration of host transcriptomic data further revealing the impact on host physiology, immune responses, and disease susceptibility [43,44,45]. Moreover, so-called ‘dysbiosis’ or shifts in the microbiome trigger changes in the host proteome [46], stimulating host cells to produce or modify specific proteins, thus offering further insights into host–microbiome interactions [47,48,49]. Lastly, at the metabolomic level, the microbiome significantly contributes to the host metabolome by producing various metabolites through microbial metabolism. These microbial metabolites directly affect host physiology, nutrient metabolism, and signaling pathways [50,51,52]. 

The integration of multi-omics layers offers a powerful and comprehensive approach to understanding human health and disease in personalized and precision medicine. Each omics layer contributes unique insights, and the interplay between them provides a more holistic view of the complex molecular mechanisms underlying various conditions. While the health condition of an individual will greatly dictate the relative importance of a single or combination of omics layers and testing frequency, we propose after an individual/patient’s multi-omics profile has been completed, emphasis should be placed on more sensitive layers such as the transcriptome and metabolome. Moreover, the microbiome’s responsiveness to nutrition (i.e., GM) and lifestyle factors makes it another important layer to sample more frequently [38,53], as it holds the potential for tailored interventions to optimize individual health outcomes.

Such a set up may prove especially fruitful in the case of preventive care, where omics data can be used as a more sensitive barometer for precision medicine by providing dynamic insights into an individual’s health status and disease risk. By analyzing metabolomic data over time, healthcare providers can identify early metabolic disturbances or deviations from a healthy baseline, enabling early detection of potential health issues or disease risk. For example, certain metabolomic profiles may indicate metabolic imbalances associated with prediabetes, cardiovascular disease, or other chronic conditions [54,55,56]. Detecting these early warning signs allows for timely interventions, such as lifestyle modifications, dietary changes, or personalized treatments, to prevent or mitigate the progression of diseases before they become clinically apparent. Frequent sampling and analysis of GM data can also provide insights into how changes in diet, exercise, medications, or other lifestyle factors shape the composition and function of the microbiome. For example, once in adulthood, the main moderator of the relationship between us (the host) and the GM is diet [57,58,59]. Microbes in lower regions of the gastrointestinal tract can produce energy and signaling compounds such as short-chain fatty acids created by bacterial fermentation of dietary fibers and resistant starch in the colon. These and other microbial agents may enable host–cell signaling through various surface receptors on the luminal side of the gut wall [60], playing a role in systemic inflammation [61], hepatic metabolism [62], and even cognition [63]. Ultimately, the production and modification of these metabolites and their interactions with the host are largely influenced by host diet and feeding behaviors. Important to note, one of the major challenges for human microbiome research is understanding where the GM fits within the framework of causality and what its role is in driving a particular health condition [64,65,66], specifically whether an altered GM is a cause or an effect of a particular disease. Metaphorically, the GM often assumes a co-pilot role with the host [67]. Indeed, a perturbed GM could be engaging in compensatory actions aimed at conferring host health benefits. Therefore, microbiomic data need to be placed within the context of the host and should be complemented with other, host-centric omics layers such as transcriptomics and metabolomics.

Another important consideration is the timing of sample collection(s). A common protocol in clinical settings is overnight fasting before blood assays, designed to minimize the immediate effects of dietary intake for accurate and reliable results. However, at a systems biology level, this approach fails to account for the circadian variations that provide critical insights into an individual’s health and biological status. Indeed, well-documented circadian fluctuations in clinical outcomes such as blood pressure, heart rate, and glucose levels underline the significance of these rhythmic patterns [68,69]. These rhythms, which may manifest on a daily (circadian), monthly (infradian), or even hourly (ultradian) basis, are consistently observed across transcriptomic, proteomic, and metabolomic levels [70,71,72]. Notably, Zhang et al. (2021) reported in their global transcriptome analysis of 12 organs that 43% of all protein-coding genes exhibited circadian rhythms in transcription, often varying organ-specifically and coinciding with shifts in light–dark cycles [73]. While the exploration of multi-omics spatiotemporal relationships—often referred to as the ‘fourth dimension’—remains an emerging field [74], integrating circadian medicine into multi-omics approaches offers a more comprehensive understanding of biological systems. For instance, circadian rhythms significantly influence an individual’s response to drugs and therapies by affecting the pharmacokinetics processes of absorption, distribution, metabolism, and excretion [75].

Recognizing the dimension of time is pivotal in biomedical analyses, particularly with the advent of less costly and longitudinally rich data sources such as wearable devices. Actigraphy, for instance, has provided substantial insights into behavioral patterns that interact with all omics layers, influencing or being influenced by them. The integration of extensive, longitudinal multi-omics data from sources such as the UK Biobank [76], which encompasses approximately 500,000 participants, has profoundly deepened our understanding of how behavioral patterns impact health, underscoring the critical need to consider temporal dynamics in omics analyses [77,78,79]. For instance, studies have shown that polygenic risk scores for low relative amplitude—an objective measure of rest–activity cycles derived from accelerometer data—are significantly associated with mood instability, major depressive disorder, and neuroticism [77]. This comprehensive approach not only enhances our understanding of biological systems but also forwards precision medicine by mapping the temporal dynamics of disease processes and treatment responses. Such insights are vital for developing tailored therapeutic interventions, as the time of medication administration can significantly affect drug pharmacokinetics, altering their effectiveness and side effect profiles. Furthermore, increased risk for diseases such as cancer, metabolic disorders, and neurological conditions frequently exhibit circadian disruptions [80,81,82], highlighting the importance of circadian data in identifying potential therapeutic targets.

The advent of wearable technology, such as devices used for actigraphy, has transformed data analysis by offering integrative, non-invasive methods for monitoring health. These devices not only facilitate the study of sleep patterns and activity cycles but also allow for the monitoring of biochemical markers, thus providing a rich dataset for integration with more traditional omics analyses. For example, the integration of continuous glucose monitoring systems with blood metabolomics has provided valuable insights into the metabolic profile changes associated with glycemic variability in Type 1 diabetes patients [83]. This success has paved the way for the development of other non-invasive, miniaturized, and cost-effective systems for the continuous measurement of biochemical analytes. Innovations include the analysis of temporal changes in sweat biomarkers [84], electrochemical measurements of synthetic miRNA strands [85], and the monitoring of thermodynamic metabolic data from the breast skin surface for early breast cancer detection [86]. Although the range of molecular analytes that can be continuously monitored on the body is currently limited to a select number of metabolites and electrolytes, the field of at-home microfluid sampling, including wearable technologies, is rapidly expanding. This area holds significant promise but is beyond the scope of this review. Readers interested in deeper exploration of these developments are encouraged to consult recent specialized reviews (see: [87,88]).

Overall, the frequent sampling and analysis of transcriptomic, metabolomic, and microbiomic data offer a more sensitive approach to preventative medicine. These omics layers provide valuable information about an individual’s unique biochemistry and microbial ecosystem, allowing for personalized and proactive healthcare strategies that focus on prevention, early detection, and optimizing health outcomes. By leveraging these data, healthcare providers could shift from a reactive model of care to a proactive one, empowering individuals to take charge of their health and reduce the burden of chronic diseases. However, some of the major hindrances in the way of realizing this are based on forming actionable and condensed metrics informed by omics and accurate and sensitive intra-level comparators. 

## 3. Creating Omics-Informed Health Indices: Meaningful Offerings for Patients and Providers

The translation of omics layers into clinical practice presents a formidable task, demanding the distillation of extensive data into coherent metrics, disease signals, and valuable health insights. The critical objective is for healthcare providers to deliver succinct, evidence-based guidance that resonates with both patients and physicians. Essential to this effort is the preservation of simplicity and transparency, forming the bedrock of the credibility of the information conveyed. Despite physicians’ expertise, the interplay among diverse omics layers and their intricate relationships can be overwhelming. The situation is further compounded by the time limitations within medical consultations, which often occur intermittently [89]. We contend that the successful integration of multi-omics analysis into clinical practice, yielding actionable patient insights, hinges upon the ability to streamline integrative assessments into meaningful health modules or indexes. While we are limited on a formalized path to achieve this, a theoretical workflow is presented in Figure 3. This index or metric must also be adaptable for continuous engagement, supporting individuals as they endeavor to ameliorate disease states or enhance their well-being throughout their lifespan.

In recent years, AI has emerged as a promising tool for managing and translating multi-omics data into clinical care. AI’s robustness is derived from its ability to recognize patterns and relationships within extensive, multidimensional, and multimodal datasets, rather than foreseeing challenges in a traditional sense [90,91]. Such capabilities enable AI systems to, for instance, condense a patient’s comprehensive medical record into a single numerical representation indicative of a probable diagnosis or transform image pixels into coordinates pinpointing tissue pathology [92]. As AI systems are dynamic [93], they can continuously learn and adjust with the introduction of new data [94], showcasing a form of adaptability.

Machine learning (ML), a prominent subset within the domain of AI, holds significant promise for multi-omics data analysis. ML tools employ statistical methodologies and algorithms to navigate the complexities of data, such as parsing, classification, and pattern identification [95]. This standard workflow in ML encompasses stages like data preprocessing, algorithm and model selection, training, testing, tuning, and validation [96]. However, ML application in multi-omics analysis presents a fundamental challenge known as the ‘curse of dimensionality’. This issue arises when datasets possess a disproportionately large number of features (*f*) compared to available samples (*n*) [97]. Contrary to conventional paradigms, an optimal ML setting requires more samples but fewer features (*f* << *n*) [98]. Such high-dimensional environments can lead to data sparsity, complicating accurate inferences and predictions without ample data points. As a countermeasure, strategies focus on both augmenting sample numbers and diminishing feature dimensions [98]. These tactics encompass feature selection, which maintains a subset of the initial features, and feature extraction, which consolidates and transfigures features into a more condensed set. Once the data are reshaped, the focus in multi-omics analysis pivots to integration. The integration of multi-omics data can be achieved through three distinct strategies: early, intermediate (middle), and late integration. These strategies have been alternately termed bottom-up and top-down approaches [99,100]. The early integration strategy, although straightforward, involves merging diverse features from each omics layer into a consolidated dataset. This method, however, is not exempt from the ‘curse of dimensionality’ due to the vastness of the feature space. On the other hand, the late integration method conducts independent analyses within each omics layer, culminating in a subsequent fusion of results. Intermediate integration offers a compromise, utilizing various transformation techniques to produce intermediary data representations. This approach effectively captures supplementary information inherent in each omics layer and accommodates novel interactions across these strata [101].

The journey towards effective multi-omics integration has led to the development of numerous models [102,103,104,105]. A central challenge remains the interpretability of results, particularly understanding the individual versus combined omics interactions [102,103,106,107]. In multi-omics analyses, the aim is to form networks between layers, potentially evolving into modules based on health or disease signatures. These modules, if demonstrating consistent associations, can be reinforced over time [108]. Clinical studies suggest that modules of disease-associated genes are valuable for inferring biomarkers and therapeutic targets [109]. Yet, a gap remains between multi-omics signatures and practical, understandable metrics for providers and patients. Notably, biomarkers often have high specificity to particular study cohorts or disease states, which limits their broader clinical application and understandability. This challenge ties back to issues of transparency and explainability, as highlighted by the Public Health Genomics foundation [110]. Deep learning has shown promise in directly translating multi-omics data into actionable insights. Notably, graph convolutional networks, which utilize convolutional layers to extract pivotal features from input graphs, show potential, especially for multi-omics [111]. For example, the Multi-Omics Graph cOnvolutional NETworks (MOGONET) employs a supervised approach for multi-omics integration, aiming to pinpoint disease-related biomarkers and predict outcomes [112]. Another tool, DeepOmix, introduces an interpretable multi-omics deep learning framework for cancer survival analysis [113]. It stands out due to its capability to integrate multi-omics data using prior biological knowledge. This knowledge can be in the form of tissue networks, gene co-expression networks, or known biological signaling pathways. While it is designed for prognosis prediction, its authors believe it can also predict other clinical outcomes such as cancer subtypes, stages, or drug responses. To date, its application has been primarily in stomach adenocarcinoma prognosis prediction [114]. Although other tools in oncology exist, their adoption remains limited [115,116]. 

Despite over a decade of significant emphasis and the development of numerous tools, the integration of AI into clinical practice remains limited. Many AI healthcare products are still in their design and developmental phases [90]. The challenges of interpretability, usefulness, and reliability largely contribute to these constraints [117]. In light of these challenges, we propose a potential path forward: prioritizing health indexes or modules derived from comprehensive multi-omics analysis and integration. This approach aligns with the concept of modularity, emphasizing interpretable elements that enrich model understanding [118]. Such modularity can span beyond molecular interactions, capturing various disease-relevant variables, from symptoms to environmental factors. For instance, the co-occurrence of symptoms such as wheezing, cough, and dyspnea could be categorized into an “asthma” module. Leveraging ML techniques, one can deduce combinations of biomarkers from these modules, offering valuable genetic, diagnostic, and therapeutic insights. Additionally, tools that identify key genes within modules can be invaluable. Genes situated at central nodes denoting high interconnectivity are of particular significance [119]. This interconnectedness can encompass a range of factors, from genetic variations to routine clinical indicators such as age and environmental influences.

In the evolving landscape of healthcare, the concept of health trajectories is gaining prominence, especially given the preventive capabilities of AI systems [90]. Operating with intentionality and adaptability, AI systems are well-suited to predict and address emerging health issues. The determination of impactful health improvements hinges on individual health objectives, specific conditions, risk factors, and the potential efficacy of interventions in enhancing overall well-being. Crucially, the integration of AI technologies is not intended to supplant the essential human elements in medical practice, but rather to augment them, enhancing the efficiency and effectiveness of these interactions. The future of AI innovations in healthcare rests on a deep, human-centric comprehension of the intricacies inherent in a patient journey and care pathway. This holistic understanding will drive the realization of AI’s transformative potential in healthcare, ensuring that it seamlessly aligns with the complex dynamics of patient well-being.

## 4. Digital Twins for Precision Medicine: Pioneering Personalized Health Insights

The healthcare system has been digitizing for quite some time, from electronic health records to virtual care, and now more recently, the incorporation of AI for diagnostics and the use of blockchain technologies for billing, claims, consent management, etc. Notably, Digital Twins (DTs), or digital replicas of real entities, have emerged as a promising tool in this regard. Originally introduced by NASA to address logistical challenges in aerospace production, DTs have since transitioned from manufacturing and aerospace to various digital domains, including biomedicine [120]. In this new application, they seek to mirror the life of their physical counterpart: the human body [121]. In the realm of precision medicine, DTs offer a unique solution, particularly for handling the massive datasets generated by longitudinal multi-omics analyses. At their core, the strength of DTs lies in their capacity to build predictive simulation models, reminiscent of stochastic models such as Markov chains. Implementing DT modeling within the healthcare framework, as we propose, necessitates a robust multi-omics pipeline with several key components: (1) data configuration to ensure compatibility and harmonization, (2) continuous data collection and patient monitoring, (3) bench-marked, high-throughput multi-omics assays, (4) diagnostic and prognostic modeling, and (5) patient–provider interactions informed by this modeling. The ultimate goal of this pipeline is to align the physical patient’s health trajectory with the corresponding system’s DT, as depicted in Figure 4. To realize such a pipeline, however, will require building out and refining all of the aforementioned components.

For the first component, data configuration is simple, although vital, to realizing any DT pipeline. The act of harmonizing data within and across entities is an obvious obstacle that we will not cover in depth here. Briefly, standardized data collection and transmission, sample and patient identifiers, reference standards, data structure, ability to evolve/modernize, etc., are all vital [122]. The second component is fairly straightforward, although there are two important considerations we would like to note: (1) the cadence of sampling will have to operate at a baseline level and increase should system changes indicate a greater chance of ailment or appearance of disease to provide more sensitive feedback (i.e., online- vs. offline- vs. backbox-twins) [123]. Establishing such a cadence is currently not straightforward and will require biological coherence; (2) Beyond the samples used as inputs for multi-omics analyses, other data will be crucial as well, including patient metadata, previous history, and perhaps data from medical sensors and wearable devices such as temperature, heart rate, blood pressure, insulin, and other fitness activities (depending on the case and application). Together, these analyses are amenable to *n*-of-1 modeling, particularly considering these models are designed to tailor medical treatment and understand individual responses to therapies over time. Indeed, *n*-of-1 trials provide empirical evidence about what works best for the patient based on actual trial data, while digital twins can explore a broader range of scenarios virtually, including those not yet tested in real life. The integration of these models with time-series analytics provides a robust feedback loop where the digital twin is continuously refined based on outcomes from *n*-of-1 trials. This allows for adjustments in the model’s predictions and, potentially, in the ongoing treatment regimens being tested in real-time *n*-of-1 studies. As a useful case example, continuous glucose monitoring devices are currently being used in this application [124] and attempts have been made in the realm of dietary response [125]. However, in the future, more sophisticated models will need more complex integration with the Internet of Things (IoT). Such integration will provide technical support for real-time data collection through sensors, data acquisition cards, and 2D codes, allowing continuous monitoring and feedback to optimize models and tailor interventions. For component 3, we have already discussed each omics layer, although it is important that there is use of standardized methods for reproducibility and continual quality checks, as with any laboratories used in medical diagnostics. Data collection in many ways is generally more simplified in comparison to building a digital model and subsequent predictions for biological response and intervention.

In the context of multi-omics integration and medicine, as discussed previously, effective diagnostic and prognostic modeling (component 4) is critical. Digital twins should be viewed as “systems” or “processes” that model the entire biological system and focus on specific parts to understand how they work together or against each other. Thus, DTs for a single patient should have a multitude of DT health indexes or modules, facilitating the derivation of multiple health trajectories. These trajectories, in turn, serve as personalized health benchmarks for individuals. By comparing real-time health data against these personalized targets, deviations can be detected and progress towards desired health outcomes can be assessed. In terms of systems integration, we propose forming DTs for each index/module or even each omics layer, informed by the other systems. Having multiple levels of systems resolution accomplishes several important things. First, the potential for precision medicine lies in high-fidelity modeling and in silico simulation that may enable “tuning” individuals for optimal biological “operation” and health outcomes, essentially monitoring biomarkers linked to disease risk and gauging the effectiveness of interventions or treatments. Taking one holistic outcome, such as continuous glucose monitoring, does not offer a deep “under-the-hood” view, and while powerful for informing an insulin dosing schedule, it fails to provide the breadth of etiological insight plasma metabolomics affords. There are many challenges applying this concept to medicine due to its more chronic and multifactorial nature. Indeed, much of the “tuning” would encompass drug therapy, diet, and behavioral regimens. This is particularly true when considering which side of a disease a healthcare provider is working on, i.e., preventative vs. curative. Such a model would need to be recalibrated periodically so that it continues to represent the index or system as it changes over the lifespan of its operation [126]. As is the case with the current multi-omics study landscape, there is the issue of missing data, whether that is for a single omics layer or an entire time point. In these scenarios, bootstrapping might need to be applied for missing data, although this should not be overutilized. Difficulty in realizing effective calibration will likely reside in the sparse nature of multi-omics profiling at present [126]. In the future, there will be a need for low-burden, high-frequency, feasible sample collection. Many promising options are emerging or are being refined, including dry blood spots [127], saliva [128], and discrete, low-burden fecal swab/wipe collections [129].

The last component, patient–provider interactions (component 5), relies on healthcare integration and will require multi-scale/personnel knowledge and skill. Indeed, training an appropriate workforce to handle this will be paramount, including programmers, healthcare workers, etc. The healthcare provider interacting with the patient will be one part of the team. As the agent delivers care in our current healthcare model, it is not reasonable for them to have a deep understanding of DTs and much of the information supplied to them must be digestible and transferrable to the patient. As an example, from the Swedish Digital Twin Consortium, using RNA, researchers formed DTs to “treat”, in silico, a patient with thousands of drugs to find the optimal compound with the best effect for the patient [130]. This information gives a provider a powerful tool with information (i.e., medication) they can understand and treat. In our model, we envision a more holistic approach with the formation of the “Master DT” which informs an overall health score. As with the goals of the Swedish Digital Twin Consortium, such information can provide tangible and effective strategies that would include diet, exercise, nutritional supplementation, etc. 

As an emulation of an individual’s biological characteristics, merging DT technology with ML-driven analysis of patient data offers a robust approach to unraveling the intricacies of wellness and disease. By creating virtual representations of health statuses, DTs have the potential to meticulously monitor changes over time. The utility of DTs spans various impactful health improvements, contingent on specific contexts and individual circumstances. The immediate enhancements already being realized in some instances include managing chronic diseases (e.g., diabetes), achieving and sustaining healthy weight, optimizing nutrition and physical activity levels, and managing stress and mental well-being. Personalized monitoring of biomarkers linked to disease risk and gauging the effectiveness of interventions or treatments [130] are already showing great promise, although they pose many important issues in data management.

## 5. Leveraging Blockchain Technology for Effective Multi-Omics Data Management

Longitudinal multi-omics assays and DTs present many challenges for data management, including trusted data provenance, auditability, traceability, and security, to name a few [131]. Importantly, health data from an individual and DTs create data related to personal information, transactions with multiple personnel and facilities, time and location logs, etc., which are not secure or tamper-proof [132]. Moreover, because of the number of transactions involved, maintaining data immutability and provenance can be difficult. Since multidisciplinary teams are heavily involved in healthcare, the interaction between the teams, workflows, and progress needs to be monitored in a trustworthy manner. Each collaboration activity that occurs between different health providers, engineers, and case managers must be documented in such a way that it ensures transparent history monitoring, traceability, privacy, trust, and security. Blockchain technology can better manage many of these issues.

The rise of blockchain technology as a responsible and transparent mechanism to store and distribute data is paving the way for new potential methods of solving serious data privacy, security, and integrity issues in healthcare [131]. A blockchain is a distributed tamperproof database, shared and maintained simultaneously by multiple parties on multiple systems. Blockchain technology is identified as a distributed ledger technology for peer-to-peer (P2P) network digital data transactions that may be publicly or privately distributed to all users, allowing any type of data to be stored in a reliable and verifiable way [133]. Blockchain allows the exchange of timestamped events and notifications which are permanently stored in a secure and tamper-proof ledger [134]. Another main concept of the blockchain is the smart contract, a legally binding policy that consists of a customizable set of rules under which different parties agree to interact with each other in the form of decentralized automation [135]. This technology enables transparency and eradicates the need for third-party administrators or intermediaries, which can present hindrances relating to delivering effective healthcare. In particular, the decentralized nature of blockchain technology allows patients, doctors, and healthcare providers to share the same information quickly and safely by exchanging their data using a blockchain network [131]. Figure 5 conceptualizes a private blockchain system in healthcare data management.

Some blockchain-based personal health information management systems have been proposed, including eHealthChain. This chain manages health data originating from medical IoT devices and connected applications [136]. Similar approaches have been proposed and have shown utility in sharing genomic data [137]. All management systems require that data sharing be done with consideration of all aspects related to patient privacy and consent, as well as the proper use of the data that must be regulated according to national and international regulations. In many scenarios related to research and medical entities, private or permissioned blockchains make the most sense since data privacy is a key issue, and there is a need to restrict access to the blockchain. Private blockchains grant specific rights and restrictions to participants in the network who are required to obtain an invitation or permission to join [138]. This is especially true considering compliance with various governing bodies and regulations including the Health Insurance Portability and Accountability Act (HIPPA) in the United States and the General Data Protection Regulation (GDPR) in the European Union. A key feature of blockchain technology in healthcare is interoperability, facilitating the exchange of health-related information, such as electronic health records, among healthcare providers and patients so that the data can be shared throughout the environment and distributed by different hospital systems [139]. Interoperability enables providers to securely share patient medical records (given patient permissions to do so), regardless of the provider’s location and the trust relationships between them [140]. A private or consortium (consisting of multiple trusted organizations) blockchain can be implemented in healthcare settings to enhance data security, integrity, and efficiency in managing and sharing sensitive test results and related information. A medical entity could leverage a private blockchain as follows:Access Control: The entity sets up a private/consortium blockchain network and invites trusted participants, such as healthcare providers, medical professionals, and researchers, to join the network. Access to the blockchain is controlled through authentication and authorization mechanisms, ensuring that only authorized users can participate in the network.Data Privacy and Security: Test results and sensitive patient information are stored in encrypted form on the blockchain, ensuring that only authorized users can access and view the data. Additionally, the blockchain’s immutable nature ensures that data cannot be altered or tampered with once recorded, enhancing data integrity.Recording Test Results: Test results are recorded as transactions on the private blockchain. Each test result transaction contains relevant information, such as the patient’s identity (protected by cryptographic keys), the test type, the timestamp, and the results themselves.Data Sharing and Consent Management: Authorized participants can access and share test results with the patients or other healthcare providers involved in the patient’s care. Patients can provide consent for sharing their test results, and the blockchain’s transparency allows them to track who accessed their data and when.Auditing and Compliance: This enables the real-time auditing of test results and data access, providing an immutable record of all transactions on the network. This feature helps the laboratory testing company to maintain compliance with data protection regulations and healthcare industry standards.Interoperability: Private/consortium blockchains can be designed to be interoperable with existing healthcare systems and databases, facilitating seamless integration of laboratory test results into electronic health records or other medical records systems used by healthcare providers.Smart Contracts: This enables the utilization of smart contracts, which are self-executing contracts with predefined rules and conditions. Smart contracts can automate certain processes within the laboratory testing workflow, such as sending notifications to patients or healthcare providers when test results are ready, or triggering specific actions based on predefined criteria.

Another important issue relates to patients’ data ownership and protection. Patients should have control and ownership over their personal health data, as the data contain sensitive information. This should also extend to an individual’s DT, as this is essentially their in silico “biological property”. In a private blockchain system, patients would have the ability to manage their data and grant or revoke permissions for sharing. Through smart contracts or other mechanisms, patients can provide consent for specific data-sharing activities. For example, a patient might choose to share their deidentified genomic data with a drug company for research purposes or with a research institution for a clinical trial. Such secured sharing events have been piloted in cancer patients with deidentified patient data derived from standard-of-care imaging, genomic testing, and electronic health records in a blockchain environment [141]. Additional considerations for patient sharing could also include specifying the duration of data sharing and setting time limits on how long their data can be used by other organizations. However, it is essential to note that data sharing in healthcare and research must comply with applicable data protection and privacy regulations (e.g., HIPAA and GDPR). These regulations govern how patient data can be used, shared, and protected, even when it is deidentified.

Tokens could be generated and used in a private blockchain system to facilitate data sharing and provide patients with more control over their deidentified data. In such a setup, tokens could represent units of value or access rights to the patient’s data. When outside entities, such as drug companies, doctors, or researchers, want to access the patient’s deidentified data for specific purposes, they would need to obtain tokens from the patient as a form of permission. Below is a summary of how this process could work:Data Tokenization: The patient’s deidentified data is tokenized, which means it is converted into a unique digital token on the private blockchain. The token could contain information about the type of data being shared, the duration of access, and any restrictions or conditions set by the patient.Token Exchange: When an outside entity wants to access the patient’s data, they must request the corresponding tokens from the patient. This request could be made through a smart contract on the blockchain, which automates the exchange process.Patient Consent: The patient reviews the request and decides whether to grant or deny permission to the outside entity. If they agree, they transfer the required tokens to the requesting entity.Data Access: Once the outside entity possesses the necessary tokens, they can use them to access the patient’s deidentified data on the private blockchain. The data can be shared securely and transparently, with the patient’s permission recorded on the blockchain.Token Validation: The blockchain ensures that the tokens are genuine and valid for the specific data access requested. This validation mechanism prevents unauthorized access to the patient’s data.Data Usage Tracking: The blockchain can track how the tokens are used, providing an auditable record of data access and usage. This transparency enhances data governance and accountability.

By implementing token-based data access and permission mechanisms, patients could have more direct control over who can access their deidentified data and for what purposes. It also allows patients to manage their data-sharing preferences more effectively. For example, patients can grant temporary access for a specific research study or revoke access at any time if they change their minds or if the data’s intended use no longer aligns with their preferences. Tokenization in a private blockchain system adds an extra layer of privacy and security to patient data, ensuring that data sharing occurs with the patient’s informed consent and that access is controlled and auditable. There could also be potential financial incentives. For instance, patients could be rewarded with tokens or other incentives for sharing their data. These incentives could be in the form of discounts on healthcare services, contributions to health savings accounts, or even direct financial compensation.

Implementing blockchains in healthcare data management comes with its own challenges. These include scalability, standardization, regulatory compliance, and addressing the technical complexities associated with integrating blockchains with the existing healthcare infrastructure. In summary, leveraging blockchain technologies in the context of multi-omics data collection can provide patients with ownership, control, security, and privacy over their health information. Blockchain has the potential to enhance data protection, consent management, and interoperability, thereby empowering individuals and enabling secure sharing and utilization of multi-omics data for personalized healthcare.

## 6. Prospects and Directions for Multi-Omics in Precision Medicine

As the field of multi-omics continues to advance, its potential to revolutionize healthcare through precision medicine becomes increasingly apparent. This section explores the prospects and directions that could shape the next decade of multi-omics research and application.

One of the most exciting emerging trends is the integration of AI and ML with multi-omics data analysis. AI and ML algorithms can handle the complexity and volume of multi-omics data, identifying patterns and correlations that would be challenging for human analysis alone [142]. These technologies are expected to enhance the predictive power and precision of multi-omics studies, leading to more accurate disease models and personalized treatment plans. Another trend is the increasing focus on real-time longitudinal data collection. Wearable technologies and advanced sensors are making it possible to gather continuous multi-omics data from individuals, providing a dynamic view of their health status. This real-time monitoring could enable the early detection of disease and more responsive healthcare interventions.

The development of more sophisticated bioinformatics tools and platforms will be crucial for the future of multi-omics. These tools will need to facilitate the integration and analysis of diverse omics datasets, from genomics and transcriptomics to proteomics and metabolomics. Advances in computational power and storage, coupled with cloud computing, will also support the handling of large-scale multi-omics data. 

Additionally, the application of blockchain technology for data management in multi-omics offers a promising avenue for ensuring data integrity, security, and patient privacy. Blockchains can provide a transparent and tamper-proof system for storing and sharing multi-omics data, which is essential for fostering trust and collaboration in the healthcare community.

The future of multi-omics will likely see increased interdisciplinary collaboration, integrating insights from fields such as bioinformatics, systems biology, computational biology, and clinical research. Collaborative efforts will be essential for tackling complex biological questions and developing comprehensive models of disease. Interdisciplinary research will also be key in translating multi-omics findings into clinical practice. For instance, collaborations between data scientists and clinicians can help bridge the gap between complex multi-omics data analysis and practical healthcare applications, ensuring that the insights gained are both actionable and clinically relevant.

Despite its promise, multi-omics faces several unresolved challenges that must be addressed to fully realize its potential. One significant challenge is the standardization of data collection and analysis methods across different omics layers and research institutions. Standardized protocols and quality control measures are essential for ensuring the reliability and reproducibility of multi-omics studies. Another challenge is the need for robust statistical methods to handle the high dimensionality and complexity of multi-omics data. Developing more sophisticated algorithms and statistical models will be crucial for accurately interpreting these data and drawing meaningful conclusions.

The applications of multi-omics in healthcare are vast and varied. In the near future, we can expect multi-omics approaches to play a central role in the development of personalized medicine. By integrating genetic, transcriptomic, proteomic, and metabolomic data, healthcare providers will be able to tailor treatments to the individual characteristics of each patient, improving outcomes and minimizing side effects. Multi-omics also holds promise for advancing our understanding of complex diseases, such as cancer, cardiovascular diseases, and neurodegenerative disorders. By elucidating the molecular mechanisms underlying these conditions, multi-omics studies can identify new biomarkers for early diagnosis and novel targets for drug development.

Over the next decade, we predict significant developments in the integration of multi-omics with other emerging technologies. For instance, the combination of multi-omics with digital twin technology could enable the creation of personalized virtual models of patients, allowing for in silico testing of treatments and interventions before they are applied in real life. Furthermore, advancements in multi-omics data visualization tools will improve the ability of researchers and clinicians to interpret complex datasets, facilitating the translation of multi-omics insights into clinical practice.

The future of multi-omics is bright, with numerous emerging trends, technological advancements, and interdisciplinary approaches poised to drive the next wave of innovation in precision medicine. Addressing the current challenges and exploring potential applications will be crucial for unlocking the full potential of multi-omics research. Continued investment in this field, coupled with collaborative efforts across disciplines, will ensure that multi-omics remains at the forefront of scientific and medical progress.

## 7. Conclusions

The integration of multi-omics analysis into healthcare stands as a pivotal step towards the realization of precision medicine’s potential. With approximately 14 billion lab tests ordered annually, clinical laboratories inform an estimated 70% of medical decisions, underscoring the importance of accurate and comprehensive data [143]. This review navigates through the challenges and opportunities inherent in transforming the clinical data landscape, identifying critical components across the translational spectrum. The formulation of health indices through AI integration, the role of AI in predictive health trajectories, and the amalgamation of advanced statistical modeling with DTs demonstrate the innovative potential to revolutionize healthcare delivery. However, these advancements necessitate addressing significant hurdles, including the creation of actionable and condensed metrics derived from omics data and the sensitive establishment of intra-level comparators. Amidst these advancements, the ethical integration of technology underscores the preservation of the human touch within medical practice. The evolution of healthcare requires a profound comprehension of patient journeys and care pathways, ensuring the harmonious alignment of AI’s transformative capacity with the intricacies of individual well-being. The integration of blockchain technology further emphasizes the need for secure and private data management, providing individuals with ownership and control over their multi-omics information. As the journey towards personalized medicine continues, the potential for AI-driven health modules, combined with blockchain’s security, provides a roadmap to a future where healthcare is proactive, precise, and patient-centric.

## Figures and Tables

**Figure 1 biomedicines-12-01496-f001:**
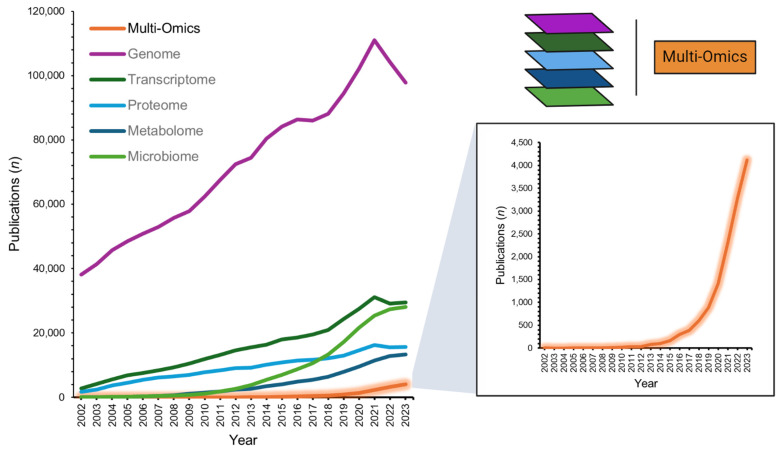
Number of multi-omics publications from 2002-2023 (PubMed database search: 31 December 2023). Data extracted from search terms [All Fields]: ‘multi-omics’; ‘genome’; ‘transcriptome’; ‘proteome’; ‘metabolome’; ‘microbiome’.

**Figure 2 biomedicines-12-01496-f002:**
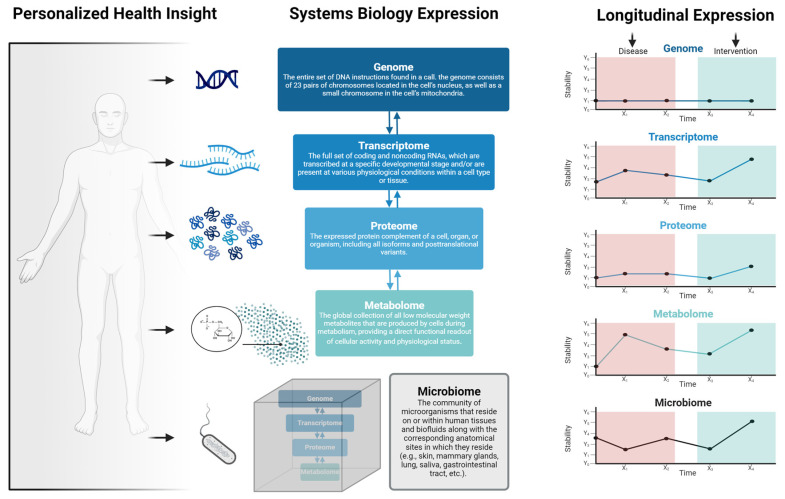
Overview of personalized medicine viewed through the lens of systems biology. Personalized health insights are derived from, and are interdependent on, layers including the genome, transcriptome, proteome, metabolome, and microbiome. Importantly, the responsiveness of each of these layers varies depending on several factors such as environmental exposures, social and behavioral activities, disease states, or health/medical interventions.

**Figure 3 biomedicines-12-01496-f003:**
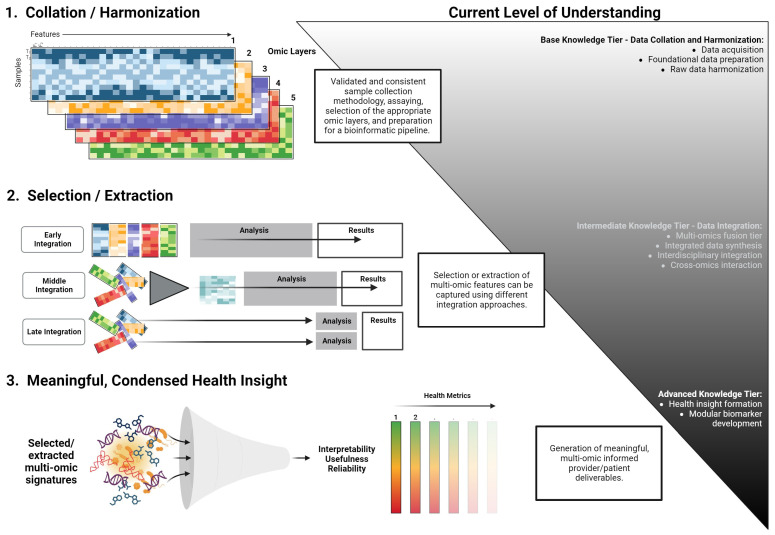
A conceptual workflow for the development of modular multi-omics health metrics. This framework outlines the transition from data acquisition and harmonization to the derivation of actionable health insights. The process is categorized into base, intermediate, and advanced knowledge tiers, emphasizing the systematic integration and interpretation of multi-omics data.

**Figure 4 biomedicines-12-01496-f004:**
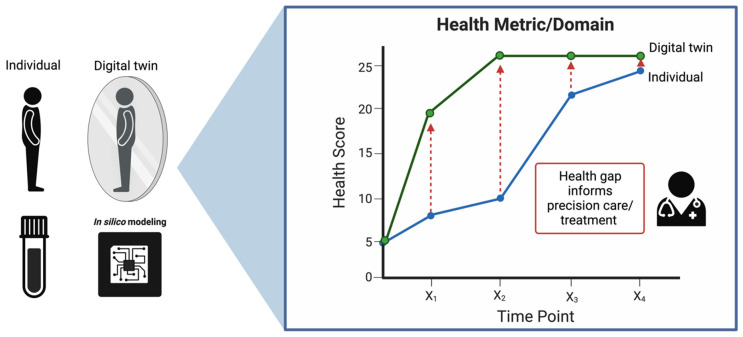
A representation of the Digital Twin framework in healthcare. On the left, an individual is mirrored by their digital counterpart, with ‘in silico’ modeling providing the computational foundation for this digital representation. On the right, the health trajectory of both the individual and their digital twin are charted over time, highlighting potential gaps in health metrics that can inform precision care and treatment strategies.

**Figure 5 biomedicines-12-01496-f005:**
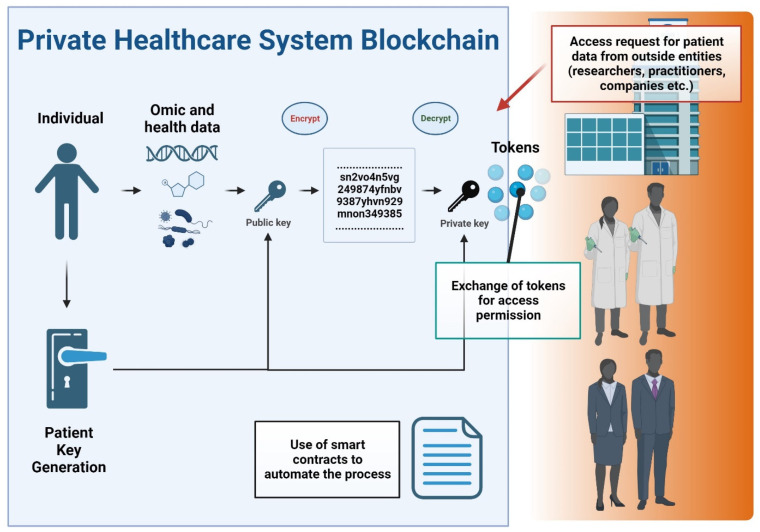
Flowchart of a private blockchain system in healthcare data management. Key components and processes are depicted as follows: The flowchart initiates with the input of patient data. Patient data undergo tokenization, converting sensitive information into secure data tokens. This process enhances privacy by ensuring that actual data elements are not directly exposed within the blockchain. External entities, such as researchers, healthcare professionals, and pharmaceutical companies, are shown requesting access to patient data. The mechanism for granting data access is depicted via a token exchange process. The use of smart contracts is highlighted, showing how they automate the decision-making process for data access, based on predefined criteria and permissions. In this process, patients actively manage their data-sharing preferences, granting or revoking permissions, which underscores patient autonomy in data management.

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
