# Peer review of "Navigating Challenges and Opportunities in Multi-Omics Integration for Personalized Healthcare"

_biomedicines, 2024, doi:10.3390/biomedicines12071496_

Round 1

Reviewer 1 Report

Comments and Suggestions for Authors

The present review Biomedicines-2990277 article, entitled "Navigating Challenges and Opportunities in Multi-Omics Integration for Personalized Healthcare", by Alex E. Mohr et al, refers to the role of multi-omics in providing comprehensive insights into complex biological systems, representing a transformative force in health diagnostics and therapeutic strategies. However, it is well known that several challenges are evident when merging varied omics data sets and methodologies, interpreting vast data dimensions, streamlining longitudinal sampling and analysis, and addressing the ethical implications of managing sensitive health information. Thus, this review evaluates these challenges while spotlighting pivotal milestones: the development of targeted sampling methods, the use of artificial intelligence in formulating health indices, the integration of sophisticated n-of-1 statistical models, such as digital twins, and the incorporation of blockchain technology for heightened data security.

However, in order for multi-omics to truly revolutionize healthcare, it demands rigorous validation, tangible real-world applications, and smooth integration into existing healthcare infrastructures.

The article focuses on the role of proteomics, metabolomics and microbiome, and particularly on the integration of these multi-omic layers, which offers a powerful and comprehensive approach to understanding human health and disease in personalized and precision medicine.

Moreover, the article provides a thorough account of the use of the Digital Twins (DTs) framework in healthcare, which was originally introduced by NASA to address logistical challenges in aerospace production. Along wth DTs, longitudinal multi-omic assays present many challenges for data management, including trusted data provenance, auditability, traceability, and security.

The manuscript is concisely written and well documented. An adequate number of the appropriate References are cited.

Overall, the article is of interest to the cognizant reader.

Reviewer 2 Report

Comments and Suggestions for Authors

The review paper addresses challenges and opportunities in multi-omics integration for personalized healthcare, deemed topical and timely.

Minor comments for improvement include:

  1. In the introduction, the authors note a significant increase in publications from 2022-2023 compared to 2002-2021. Request to elaborate on this with a detailed catalog of relevant publications, possibly presented in a table format.
  2. Concerns raised regarding the authors' use of both "personalized" and "precision" medicine terminologies interchangeably, suggesting the need for consistency.
  3. The authors employ AI in data analysis but lack depth in discussing the merits of using machine learning for data processing and its limitations.
  4. Suggestion for the inclusion of a section on future prospects and directions for the field.
Comments on the Quality of English Language

No comment.

Reviewer 3 Report

Comments and Suggestions for Authors

The topic is timely and of interst, but the novelty is marginal.

See, for example,

Challenges and opportunities with multi-omics integration in precision medicine

PROCEEDINGS ARTICLE published 2023 in Challenges and opportunities with multi-omics integration in precision medicine

Authors: Volodimir Olexiouk

 https://doi.org/10.58647/rexpo.23030

Comments on the Quality of English Language

No comments

Reviewer 4 Report

Comments and Suggestions for Authors

Review emphasizes the opportunities and challenges of Multi-Omics integration into concepts of personalize medicine.

I believe this review can be recommended, however, there is one major issue about review in its current form that must be addressed:

Review in its current version overlooks almost completely a sine qua non fourth dimension for each database which utilizes factor of “time” (e.g., doi: 10.1016/j.cmet.2019.06.019).

  1. Integration of circadian and longer monitoring data, such as actigraphy, into UK biobank already allowed insights into behaviour patterns, utilizing circadian data to promote health and wellness initiatives, encouraging healthy sleep habits and work-life balance, enhance performance and productivity.

  2. The authors only briefly on P.3. discuss possible consequences of shift-work upon rhythmic processes. However, timing and individual phase differences affect most of multi-omics at each level (e.g. doi: 10.1038/nrg.2016.150). There are many other benefits that are broadly discussed in the plenty of recent papers. The integration of circadian medicine into multi-omics approaches allows for a more holistic understanding of biological systems, facilitates the development of personalized medicine strategies, and sheds light on the role of circadian rhythms in health and disease. Such integration is crucial as the majority of variables in multio-omics databases express predictable fluctuations and rhythms, mostly circadian (but also infradian and ultradian) that commonly occur on transciptional (doi: 10.1016/j.xinn.2023.100380), post-transcriptional and translational (doi: 10.1016/j.cels.2018.10.014) levels. Therefore, integrating circadian medicine allows for the consideration of temporal dynamics in multi-omics data, providing a more comprehensive understanding of biological processes. Circadian rhythms also affect an individual's response to drugs and therapies (e.g. https://doi.org/10.1039/BK9781839167553-00536). Incorporating circadian data into multi-omics analyses enables the development of more precise and personalized treatment strategies tailored to an individual's circadian profile. Furthermore, many diseases, including metabolic disorders, cancer, and neurological conditions, exhibit circadian disruption. By integrating circadian medicine into multi-omics studies, researchers can uncover the molecular mechanisms underlying these diseases and identify potential therapeutic targets. Incorporating circadian data into multi-omics analyses enhances our understanding of complex biological systems and their associations, leading to insights into health and disease at a systems level.

  3. The authors mentioned briefly in abstract N-of-1 concepts / models, but not described how it can be integrated into multi-omics. As dynamical changes are at the very core of n-of-1 individual models, it is imperative to incorporate proper time-series analyses while working with such databases. This approach deals directly with chronobiology and chronomics (e.g. https://doi.org/10.1007/978-1-4020-6714-3_3).

    I believe paying proper attention to chronobiological aspects in elaboration of Multi-Omics concept is necessary and should considerably improve this manuscript.

Round 2

Reviewer 3 Report

Comments and Suggestions for Authors

This revised version is acceptable having in mind the relevance of the topic.

Consider inclusing some related and recent references:

PaCMAP-embedded convolutional neural network for multi-omics data integration. Qattous, H., Azzeh, M., Ibrahim, R., ...Al Sorkhy, M., Alkhateeb, A.Heliyon, 10(1), e23195, 2024.

Concept and solution of digital twin based on a Stieltjes differential equation. Area, I., Fernández, F.J., Nieto, J.J., Tojo, F.A.F. Mathematical Methods in the Applied Sciences 45(12), pp. 7451-7465, 2022.

Digital Twins in Healthcare: Methodological Challenges and Opportunities. Meijer, C., Uh, H.-., el Bouhaddani, S. Journal of Personalized Medicine , 13(10), 1522, 2023.

Finally, give some future directions on for integrating all those approaches.

Comments on the Quality of English Language

No comments

Author Response

Reviewer 3:

This revised version is acceptable having in mind the relevance of the topic.

Consider inclusing some related and recent references:

PaCMAP-embedded convolutional neural network for multi-omics data integration. Qattous, H., Azzeh, M., Ibrahim, R., ...Al Sorkhy, M., Alkhateeb, A.Heliyon, 10(1), e23195, 2024.

Concept and solution of digital twin based on a Stieltjes differential equation. Area, I., Fernández, F.J., Nieto, J.J., Tojo, F.A.F. Mathematical Methods in the Applied Sciences 45(12), pp. 7451-7465, 2022.

Digital Twins in Healthcare: Methodological Challenges and Opportunities. Meijer, C., Uh, H.-., el Bouhaddani, S. Journal of Personalized Medicine , 13(10), 1522, 2023.

Finally, give some future directions on for integrating all those approaches.

Response:

We thank the reviewer for their positive feedback and for highlighting the importance and timeliness of the topic addressed in our manuscript. We appreciate the suggestions for additional references. We have now included the recommended citations in relevant sections of our manuscript to enrich the discussion and provide broader context. Specifically, the references have been integrated as follows:

  1. Qattous et al. (2024) has been added to the section discussing advanced methodologies in multi-omics data integration (citation 9, line 50).
  2. Area et al. (2022) has been cited in the section on digital twins, emphasizing novel mathematical approaches for modeling (citation 91, line 358).
  3. Meijer et al. (2023) has been included in the discussion on digital twins in healthcare, highlighting methodological challenges and opportunities (citation 142, line 732).

We have also updated our future directions section on page 17, lines 735–739, and 745–749. We hope these additions meet the reviewer's expectations and enhance the comprehensiveness of our manuscript. Thank you for your valuable suggestions.

Reviewer 4 Report

Comments and Suggestions for Authors

Some improvements were made according to recommendations, manuscript can be recommended.

Author Response

Reviewer 4:

Some improvements were made according to recommendations, manuscript can be recommended.

Response:

We sincerely thank Reviewer #4 for their positive feedback and for recommending our manuscript for publication. We appreciate your recognition of the improvements made in response to the previous recommendations.